# communications
# engineering

# A general framework for quantifying uncertainty at scale

Ionuț-Gabriel Farcaș [1,4✉], Gabriele Merlo[1,4] & Frank Jenko [1,2,3]

In many fields of science, comprehensive and realistic computational models are available nowadays. Often, the respective numerical calculations call for the use of powerful super-computers, and therefore only a limited number of cases can be investigated explicitly. This prevents straightforward approaches to important tasks like uncertainty quantification and sensitivity analysis. This challenge can be overcome via our recently developed sensitivity-driven dimension-adaptive sparse grid interpolation strategy. The method exploits, via adaptivity, the structure of the underlying model (such as lower intrinsic dimensionality and anisotropic coupling of the uncertain inputs) to enable efficient and accurate uncertainty quantification and sensitivity analysis at scale. Here, we demonstrate the efficiency of this adaptive approach in the context of fusion research, in a realistic, computationally expensive scenario of turbulent transport in a magnetic confinement tokamak device with eight uncertain parameters, reducing the effort by at least two orders of magnitude. In addition, we show that this refinement method intrinsically provides an accurate surrogate model that is nine orders of magnitude cheaper than the high-fidelity model.

[1] Oden Institute for Computational Engineering and Sciences, The University of Texas at Austin, Austin, TX, USA. [2] Max Planck Institute for Plasma Physics, Garching, Germany. [3] School of Computation, Information and Technology, Technical University of Munich, Munich, Germany. [4] These authors contributed equally: Ionuț-Gabriel Farcaș, Gabriele Merlo. ✉email: ionut.farcas@austin.utexas.edu

During the past few decades, science has been revolutionized through computing. First, model-based computing ("simulation")[1] has allowed us to investigate complex phenomena that are not accessible to theoretical analysis alone, and now data-centric computing[2,3] is enabling us to more effectively explore, understand, and use large amounts of data originating from experiments, observation, and simulation. Building on these important advancements, we are now entering a new phase in which the focus lies on transitioning from merely interpretive and qualitative models to truly predictive and quantitative models of complex systems through computing.

One obstacle on the path toward predictive physics-based models is uncertainty. Whether stemming from incomplete knowledge of the given system, measurement errors, inherent variability, or any other source—uncertainty is intrinsic to most real-world problems, and this aspect needs to be included in respective modeling efforts. Accounting for, understanding, and reducing uncertainties in numerical simulations of real-world phenomena is performed within the framework of Uncertainty Quantification (UQ)[4,5]. In the present work, we assume that uncertainty enters the underlying model through a set of scalar inputs (parameters characterizing the state of the system, the initial or boundary conditions, or other aspects of the system under consideration), whose cardinality is referred to as the stochastic dimension or, simply, the dimension of the UQ problem. Another key task in predictive physics-based simulations affected by uncertainty is understanding the impact of input uncertainty in the simulation's output of interest, known as (global) sensitivity analysis (SA)[6,7]. Identifying the inputs with the highest sensitivity in turn facilitates, among other things, a posteriori dimension reduction (parameters with low sensitivities are fixed) and model simplification (redundant parts of the model can be removed).

It is clear, in principle, that UQ and SA are fundamental tasks for numerical simulations of real-world phenomena and highly desirable in a wide range of applications. In practice, however, concrete implementations of these ideas are often hampered by the substantial or even prohibitive computational cost that comes with large ensembles of model evaluations, especially in large-scale, computationally expensive problems[8–11]. This is because the uncertain inputs are generally modeled as random variables whose realizations are propagated through the underlying model to compute outputs of interest. And when a single evaluation is computationally expensive, performing large numbers of such simulations becomes prohibitive. Consequently, the tension between the level of realism of a given model and the associated computational cost tends to prevent UQ/SA studies in many cases of interest. One possible way out is to resort to (non-intrusive) reduced models[12], but typically at the expense of sacrificing – to some degree—accuracy, reliability, and predictability. Moreover, constructing reduced models in computationally expensive applications can be prohibitive in the first place due to the cost of acquiring the training data.

In the following, we will describe an alternative solution to this challenging problem. Our goal is to show that our recently developed sensitivity-driven dimension-adaptive sparse grid interpolation strategy[13,14] provides a general framework for UQ and SA studies in practically relevant science and engineering applications at scale. This includes fields like (computational) fluid dynamics[15]—which is one of the first fields in which UQ and SA became prominent—combustion in rocket engines[16], climate modeling[17], materials science[18], tsunami and earthquake simulations[11,19], computational medicine[20,21], or, relevant at the time of the writing of this article, the mathematical modeling of epidemics, in particular, the Coronavirus Disease (COVID-19)[22–24], to name just a few. Any of these examples, and many more, could be used to make the case, but here, we will focus on one of the key physics problems in fusion research, namely how to quantify and predict turbulent transport in magnetic confinement experiments[25,26].

The quest for fusion energy is based on the notion that the physical processes which power the stars (including the Sun) can be mimicked on Earth and used for electricity production. For this purpose, one creates very hot plasmas, i.e., ionized gases of up to about 100 million degrees, and places them in strong magnetic fields of doughnut-shaped devices called tokamaks and stellarators. The toroidal magnetic field forces the electrically charged plasma particles onto helical orbits about the field lines and thus establishes magnetic confinement. The latter is not perfect, however, since the resulting (strong) spatial temperature and density differences induce turbulent flows in the plasma. Quantifying, predicting, and controlling this turbulent transport is a prerequisite for designing optimized fusion power plants and is therefore considered a key open problem in fusion research. Over the last two decades or so, comprehensive and realistic simulation tools have been developed for this purpose. One of them is the GENE code[27], which has a world-wide user base and will be employed extensively in the present work. Quantifying the turbulent transport under specific experimental conditions is challenging[28], with computational costs typically ranging from $10^4$ to $10^5$ core-hours, and sometimes even significantly exceeding those numbers. The resulting outputs of interest, that is, time-averaged turbulent fluxes depend on several dozen of physical parameters which characterize the properties of the plasma as well as of the confining magnetic field. Obviously, this is a perfect example of a situation in which it is almost impossible to approach the UQ/SA problem via brute-force approaches due to the so-called curse of dimensionality[29], i.e., the exponential growth of the required computational effort with the number of uncertain inputs.

To combat or at least delay the curse of dimensionality, we can exploit the fact that real-world problems often exhibit structure. And in the context of UQ and SA, this implies that even though the number of uncertain inputs can be large, usually only a small subset of them are really important for the underlying simulation, in the sense that they produce significant variations in the output(s) of interest. In addition, assuming that nonlinear effects exist in the way subsets of uncertain inputs interact with each other, these interactions are often anisotropic, i.e., the interaction strength varies significantly. Having information about the importance of uncertain inputs and the strength of their interaction is clearly advantageous for UQ/SA since, for example, the probabilistic space in which the underlying uncertain parameters live can be sampled accordingly, thus potentially decreasing the number of required samples significantly. In general, however, this information—typically obtained via SA—is available only a posteriori, after the simulations have been performed. In contrast, our sensitivity-driven dimension-adaptive sparse grid interpolation approach explores and exploits this structure, online, via adaptive refinement.

In the present article, we will show that our structure-exploiting method enables UQ and SA in large-scale, realistic, nonlinear simulations, which goes beyond what most existing methods offer. We note that our framework is applicable to many large-scale computational science and engineering simulations that are typically computationally too expensive for standard methods but at the same time critical for decision making and other practically relevant tasks. As a representative of such as an application, we will study multi-dimensional UQ and SA in first-principles-based fully nonlinear turbulence simulations of fusion plasmas. In a realistic and practically relevant simulation scenario of turbulent transport in the edge of the DIII-D fusion

experiment with more than 264 million degrees of freedom and eight uncertain inputs, our approach requires a mere total of 57 high-fidelity simulations for accurate and efficient UQ and SA. In addition, since our method is based on interpolation, a byproduct is an interpolation-based surrogate model of the parameters-to-output of interest mapping. The obtained surrogate model is accurate and nine orders of magnitude cheaper to evaluate than the high-fidelity model.

## Results

**A framework for uncertainty propagation and sensitivity analysis in large-scale simulations**. Let $d \in \mathbb{N}$ denote the number of uncertain inputs modeled as random variables distributed according to a given multi-variate probability density, $\pi$. The input density stems from, e.g., experimental data analysis, expert opinion, or a combination thereof. The high-fidelity simulation code calculates an output of interest, which, for simplicity, is assumed here to be a scalar quantity, noting that our approach can be trivially employed in the multi-variate case as well by, e.g., treating each output component separately. UQ and SA generally require ensembles of high-fidelity model evaluations at samples distributed according to $\pi$. In our sensitivity-driven dimension-adaptive sparse grid interpolation strategy[13,14], these samples are points living on a $d$-dimensional (sparse) grid that is constructed sequentially via adaptive refinement. What enables UQ and SA at scale is our strategy for adaptive refinement.

Sparse grid approximations are constructed as a linear combination of judiciously chosen $d$-variate products of one-dimensional approximations. This linear combination is done with respect to a multi-index set $\mathcal{L} \subset \mathbb{N}^d$ comprising $d$-variate tuples $= (\ell_1, \ell_2, \ldots \ell_d) \in \mathbb{N}^d$ called multi-indices. Each multi-index $\boldsymbol{\ell}$ uniquely identifies a product defined on a $d$-variate subspace. Here, the underlying approximation operation is interpolation defined in terms of (global) Lagrange polynomials, which can be trivially mapped to an equivalent spectral projection approximation. We note that our approach is hierarchical, meaning that the results corresponding to a multi-index $\boldsymbol{\ell}$ can be reused at its forward neighbors $\boldsymbol{\ell} + \mathbf{e}_i$, where $i = 1, 2, \ldots, d$, and $e_{ij} = 1$ if $i = j$ and $e_{ij} = 0$ otherwise.

We construct $\mathcal{L}$ sequentially via our sensitivity-driven dimension-adaptive refinement procedure. In dimension-adaptivity[30,31], $\mathcal{L}$ is split into two sets, the old index set, $\mathcal{O}$, and the active set, $\mathcal{A}$, such that $\mathcal{L} = \mathcal{O} \cup \mathcal{A}$. The active set $\mathcal{A}$ contains the candidate subspaces for refinement, whereas the old index set $\mathcal{O}$ comprises the already visited subspaces. In each refinement step, we ascertain the importance of individual inputs and of their interactions to guide the adaptive process. To this end, we determine the sensitivity information—with respect to the output of interest—of all uncertain inputs in each candidate subspace for refinement by decomposing its associated variance into contributions corresponding to individual inputs and contributions corresponding to interactions between inputs. Note that when normalized by the variance, these contributions would respectively represent first-order and interaction Sobol' indices for sensitivity analysis. Our algorithm, however, employs unnormalized indices because the variance is constant and therefore does not change the ordering of these indices. Moreover, the variance in subspaces in, e.g., the latter stages of the refinement process is usually small, in which case such a division would be close to the indeterminate operation 0/0. We compare these unnormalized indices with user-defined tolerances—one tolerance for each unnormalized index—to compute a sensitivity score. The sensitivity score is an integer, initially set to zero, which is increased by one whenever a user-defined tolerance is exceeded. These tolerances can be viewed as a (heuristic) proxy for the

accuracy with which we want the algorithm to explore the directions associated with the unnormalized sensitivity indicators. Upon computing the scores for all candidate subspaces, we refine the subspace with the largest sensitivity score noting that if two or more subspaces have identical scores, we select the one with the largest sum of unnormalized sensitivity indicators. Note that for a problem with $d$ uncertain inputs, we have $2^d - 1$ indices in total: $d$ for each individual input, $d(d-1)/2$ for pairs of input interactions and so forth up the index measuring the sensitivity of the interaction of all $d$ inputs. When $d$ is small to moderate, e.g., $d \leq 15$, we can exhaustively compute and use all $2^d - 1$ sensitivity indices in each refinement step and thus prescribe $2^d - 1$ associated tolerances. For larger values of $d$, however, $2^d - 1$ is prohibitively large. We can, nonetheless, exploit that in most practical applications, it is unlikely that interactions beyond two or three parameters are important and therefore account for these interactions only in our refinement procedure. Furthermore, if the prescribed tolerances do not provide the desired accuracy, they can be sequentially decreased at no additional cost since our approach is hierarchical. We note that (global) sensitivity analysis, which is central to our refinement policy, reflects the properties of the underlying high-fidelity model, which means that our method does not depend on the specific implementation of the model. Rather, the method will explore and exploit the properties of the model with the goal of preferentially refining the important stochastic directions. For example, if there are $d = 20$ uncertain input parameters in total but only three are important and, furthermore, only four interactions are important as well, our approach will exploit this structure and construct a multi-index set having more multi-indices in the directions corresponding to the three important individual parameters and four interactions. In contrast, if all 20 inputs are important, the multi-index set will likely have a large cardinality, containing multi-indices in all 20 directions. We additionally note that our method can be trivially incorporated in the underlying simulation pipeline since it only requires prescribing the simulation inputs by, e.g., accessing the parameters/configuration file, and computing the value of the output of interest. In this way, the framework can be easily used on a wide range of computing systems, ranging from laptops to large supercomputers.

At the end of the adaptive process, the sensitivity-driven method yields (i) statistics such as the mean and variance of the output of interest, (ii) the sensitivity indices of all individual parameters and either of all interactions—if $d$ is moderately large —or of a subset of interactions, and (iii) an interpolation-based surrogate model for the parameters-to-output of interest mapping. Figure 1 depicts a visual summary of the sensitivity-driven dimension-adaptive sparse grid framework through an example with $d = 3$ uncertain inputs $(\theta_1, \theta_2, \theta_3)$, which requires prescribing $2^d - 1 = 7$ tolerances. Therein, $\theta_3$ is the most important parameter, $\theta_1$ is the second most important, and $\theta_2$ is the least important parameter. Moreover, $\theta_1$ and $\theta_3$ interact strongly. Notice that the sensitivity-driven approach constructs a multi-index set that reflects this structure.

**Simulation of turbulence in the near-edge region of fusion devices**. To demonstrate that our sensitivity-driven approach enables UQ and SA at scale, we employ it in the context of nonlinear turbulent transport in magnetic confinement devices, such as tokamaks or stellarators. This is a paradigmatic example in which UQ and SA are clearly needed but in which most standard approaches are infeasible due to the large computational cost of the associated simulations. The experimental error bars of various input parameters (such as the spatial gradients of the

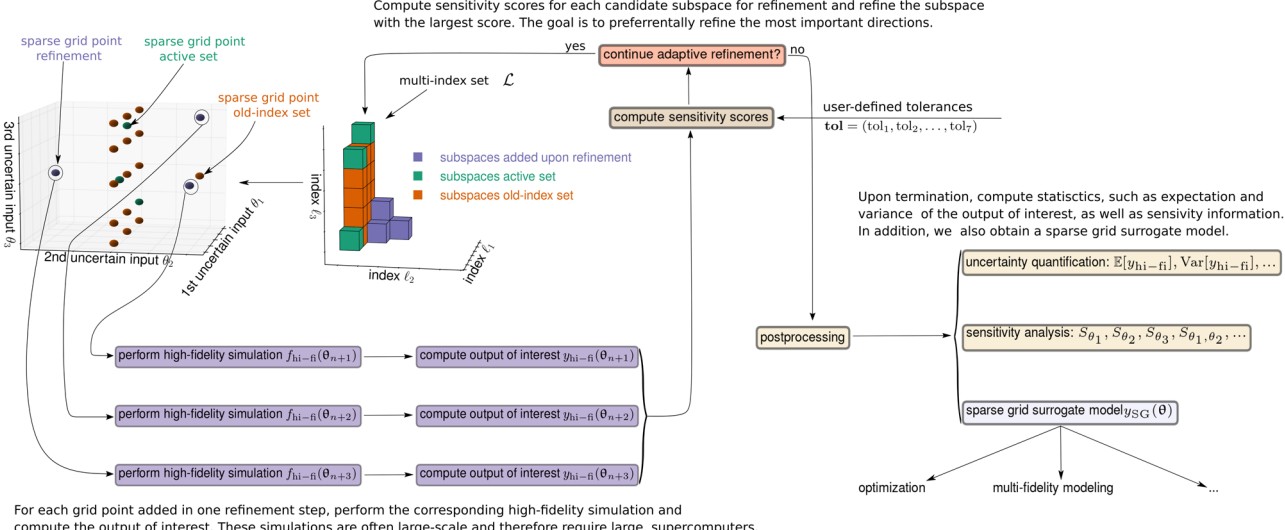

**Fig. 1 Visual illustration of the sensitivity-driven dimension-adaptive sparse grid framework for uncertainty quantification and sensitivity analysis at scale.** The framework is demonstrated in an example with $d = 3$ uncertain inputs ($\theta_1, \theta_2, \theta_3$). The goal of the sensitivity-driven approach is to explore and exploit the fact that in real-world simulations, only a subset of the uncertain inputs are important and that these inputs interact anisotropically.

density and temperature of a given plasma species) can be relatively large, on the order of a few ten percent, which makes the SA task especially valuable since understanding the impact of these uncertainties is critical. Moreover, ascertaining the impact of variations in parameters that characterize the confining magnetic field is crucial as well (e.g., for design optimization). We note that the current sparse grid framework has been employed for UQ and SA in linear simulations in fusion research (linear in the five phase-space variables that characterize the underlying gyrokinetic model) in previous efforts[13,32]. Linear simulations are often used to gain valuable insights regarding some general trends or parameter dependencies. However, here we go far beyond and consider the much more complex numerical simulations of nonlinear turbulent transport, which are necessary to make reliable quantitative predictions. These simulations represent chaotic processes in space and time, in which a large number of degrees of freedom are usually strongly coupled in a highly nonlinear fashion, and are computationally much more expensive (about four orders of magnitude higher in our case). Our main goal here is to show that our sparse grid method enables UQ and SA in such applications as well.

As a practically relevant example of nonlinear turbulent transport in magnetic confinement devices, we focus on the near-edge region of fusion experiments, which is recognized as crucial for setting the overall performance of these devices. To achieve core temperatures and densities which are sufficiently high to yield self-heated ("burning") plasmas in future power plants, it is necessary to create and sustain a region of steep gradients in the edge, known as the pedestal; see Fig. 2.

The formation of the pedestal is a very complex process, and its understanding is still incomplete at this point. A known key element in this context is plasma turbulence, which develops within the pedestal due to the very large spatial changes of density and temperature, inducing turbulent transport and hence contributing to its self-regulation. An important driver for this kind of dynamics is a plasma micro-instability called the Electron Temperature Gradient (ETG) mode, which tends to operate on sub-mm scales in planes perpendicular to the background magnetic field. Quantifying the impact of ETG turbulence on the pedestal structure is of high practical relevance, as it can aid in the design of configurations with improved energy confinement.

We consider a numerical setup modeling a specific pedestal of the DIII-D tokamak[33] and investigate the edge plasma at a normalized radius of $\rho = 0.95$. Simulations with the gyrokinetic turbulence code GENE[27] show that ETG modes are the main drivers of turbulent transport under these conditions. The employed grid in five-dimensional position-velocity space consists of $256 \times 24 \times 168 \times 32 \times 8 = 264{,}241{,}152$ degrees of freedom. The simulations are performed using 16 compute nodes, i.e., a total of 896 cores on the Frontera supercomputer at the Texas Advanced Computing Center at The University of Texas at Austin[34]. With this setup, the average run time exceeds 8000 core-hours; the smallest was about 4000 core-hours, while the largest run time exceeded 14,000 core-hours. We refer the reader to the Methods section for more details about the employed gyrokinetic model and simulation code GENE.

The experimental parameters necessary to determine the transport levels caused by ETG modes are the spatially local values of electron temperature $T_e$ (I) and density $n_e$ (II), together with their normalized inverse scale-lengths $\omega_{T_e}$ (III) and $\omega_{n_e}$ (IV). We also consider the electron-to-ion temperature ratio $\tau$ (V) and account for the effects of plasma impurities via an effective ion charge $Z_{\text{eff}}$ (VI). Basic properties of the magnetic geometry are characterized via the safety factor $q$ (VII) and the magnetic shear $\hat{s}$ (VIII). This leaves us with a total of eight uncertain parameters, summarized in Table 1, which are modeled as uniform random variables. Their respective nominal (mean) values are showed in the second column. Moreover, their left and right bounds (columns three and four) are as follows: the first two are varied by 10% around their nominal value whereas the remaining six inputs —including the two inverse scale-lengths—are varied by 20% around their nominal value; these variations reflect representative experimental error bars. We note that the larger bounds (±20%) for the inverse scale-lengths $\omega_{T_e}$ and $\omega_{n_e}$ compared to the lower bounds (±10%) used for their respective local values reflects the fact that the inverse scale-lengths are not directly available but must be computed from measured profiles. The GENE output is the electron heat flux calculated over a sufficiently long time interval to collect statistics (see the Methods section for more details). The output of interest, $Q_{\text{hi-fi}}$, is the time-averaged electron heat flux across a given magnetic surface, measured in megawatts (MW). In the following, we will show that our sensitivity-driven

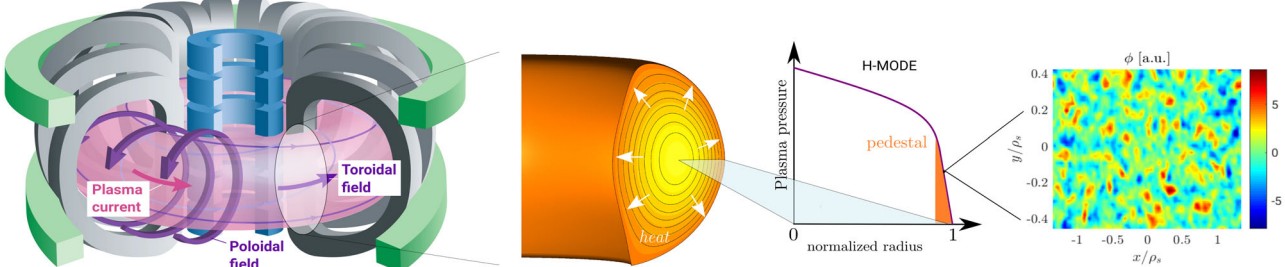

**Fig. 2 Turbulent transport in the near-edge of tokamaks.** From left to right: in the tokamak design, a hot hydrogen plasma is confined in a doughnut-like shape with the aid of strong magnetic fields (figure courtesy of EUROfusion). However, the magnetic confinement is not perfect: turbulent fluctuations driven by micro-instabilities cause heat losses from the hot core toward the colder edge. In so-called high-confinement (H-mode) discharges, one can induce the formation of a near-edge region characterized by reduced transport and steep gradients. The properties of this pedestal are influenced by the residual turbulent transport, which can be calculated, e.g., by means of the GENE code.

---

**Table 1 Summary of the eight uniform uncertain parameters considered in the numerical experiments.**

| Uncertain input parameter | Nominal value | Left uniform bound | Right uniform bound |
|---|---|---|---|
| Electron temperature $T_e$[keV] | 0.3970 | 0.3573 | 0.4367 |
| Electron density $n_e$ [$10^{19}$m$^{-3}$] | 4.4923 | 4.0428 | 4.9412 |
| Inverse electron temperature scale-length $\omega_{T_e}$ | 186.0000 | 148.8000 | 223.2000 |
| Inverse electron density scale-length $\omega_{n_e}$ | 88.0000 | 70.4000 | 105.6000 |
| Electron-to-ion temperature ratio $\tau$ | 1.4400 | 1.1520 | 1.7280 |
| Effective ion charge $Z_{\text{eff}}$ | 1.9900 | 1.5920 | 2.3880 |
| Safety factor $q$ | 4.5362 | 3.6289 | 5.4434 |
| Magnetic shear $\hat{s}$ | 5.0212 | 4.0169 | 6.0254 |

The second column shows the nominal (mean) value of the eight parameters. The corresponding left and right uniform bounds are shown respectively in the third and fourth columns.

---

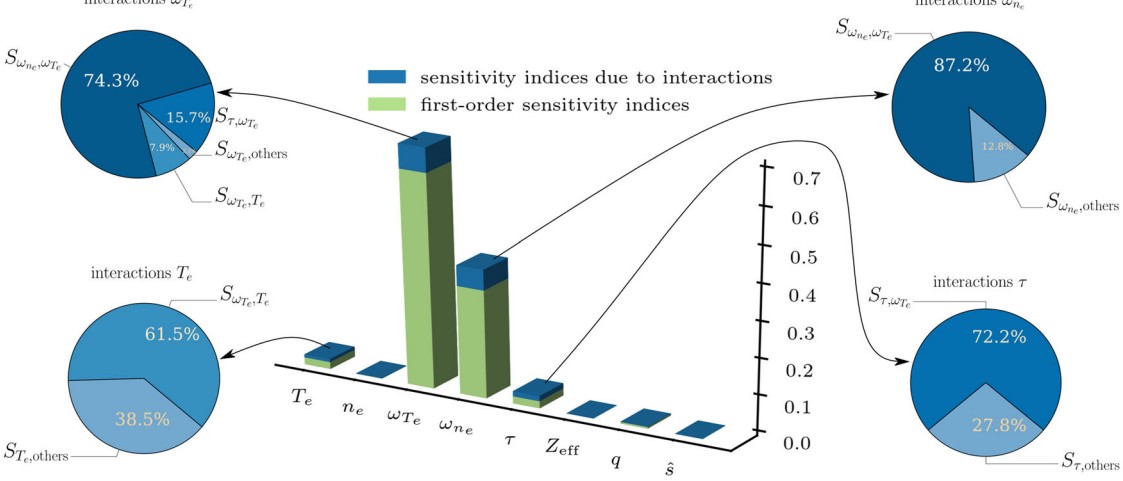

**Fig. 3 The sensitivity indices of the eight uncertain parameters.** The magnitudes of the first-order sensitivity indices and indices due to interactions are depicted in the three-dimensional bar plot. The four pie charts break down the magnitude of the interactions involving the four most important parameters, i.e., $w_{T_e}$, $w_{n_e}$, $T_e$, and $\tau$ into percentages showing the respective non-negligible interactions (the sensitivity indices corresponding to an interaction between two input parameters $\theta_i$ and $\theta_j$ are denoted by $S_{\theta_i,\theta_j}$) and all other, negligible interactions, denoted by $S_{\cdot,\text{others}}$.

---

approach enables an efficient UQ and SA in these simulations which would otherwise be impossible with standard methods.

**Accurate uncertainty propagation and sensitivity analysis at a cost of only 57 high-fidelity simulations.** The employed tolerances in our sensitivity-driven approach are $\mathbf{1}_{63} \times 10^{-4}$, which were sufficiently small for our purposes. Here, $\mathbf{1}_{63}$ denotes a vector with 63 unity entries, where $63 = 2^8 - 1$ is the total number of sensitivity indices used in each refinement step. Remarkably, our approach requires only 57 high-fidelity

simulations to reach the prescribed tolerances. This low number is due to the ability of the sensitivity-driven approach to explore and exploit that, as depicted in Fig. 3, only four (i.e., $\omega_{T_e}$, $\omega_{n_e}$, $T_e$, and $\tau$) of the total of eight uncertain parameters are important, with two parameters—$\omega_{T_e}$ and $\omega_{n_e}$—being significantly more important than the other six parameters. Furthermore, the four important parameters interact anisotropically with the other inputs. The strongest interaction occurs between the two most important individual parameters, $\omega_{T_e}$ and $\omega_{n_e}$, and the second strongest interactions is between $\omega_{T_e}$ and $\tau$. These findings are

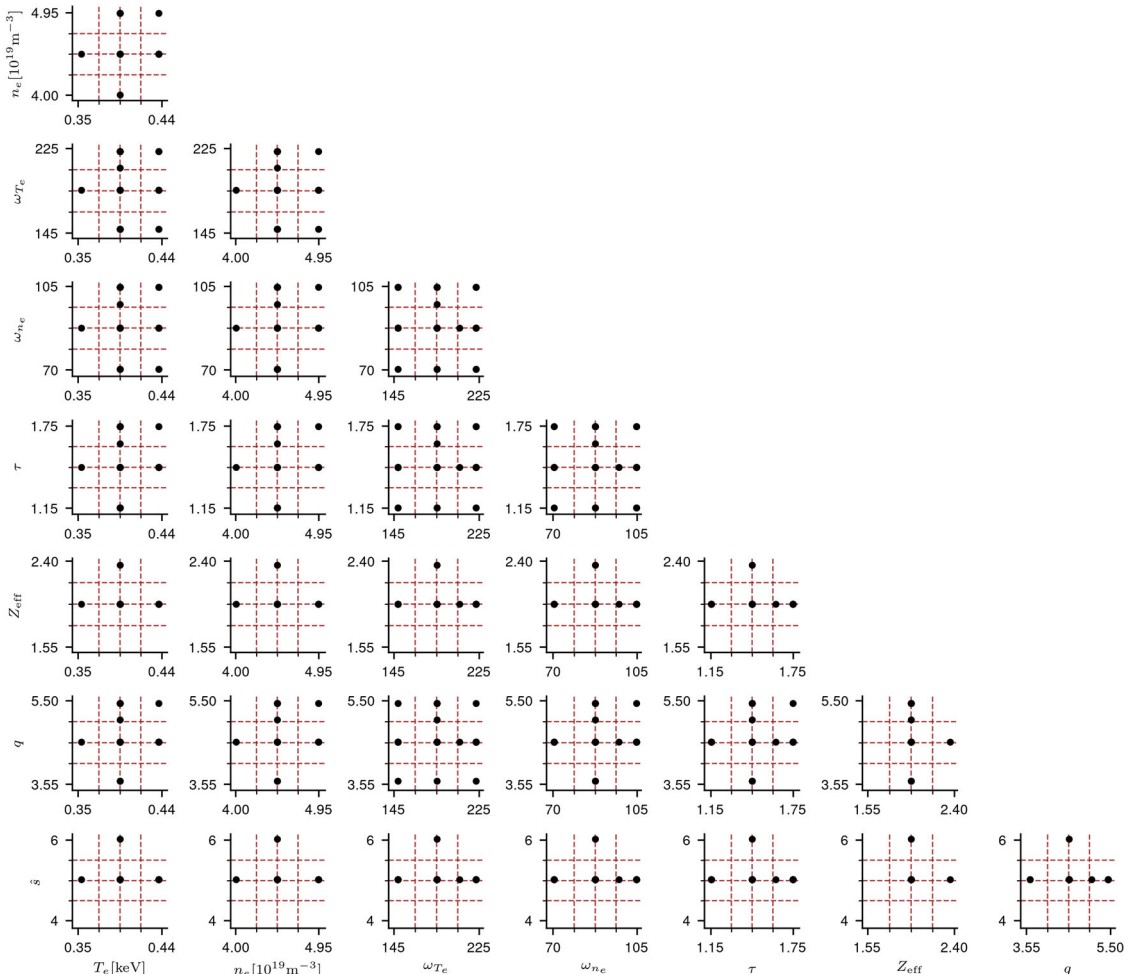

**Fig. 4 The pair-wise two-dimensional projections of the 57 sparse grid points used to explore the eight-dimensional input space.** The two-dimensional projection planes correspond to all input pairs $\{\theta_i, \theta_j\}$ where $\theta_i \in \{T_e, n_e, \omega_{T_e}, \omega_{n_e}, \tau, Z_{\text{eff}}, q\}$ and $\theta_j \in \{n_e, \omega_{T_e}, \omega_{n_e}, \tau, Z_{\text{eff}}, q, \hat{s}\}$.

consistent with generic qualitative expectations. The remarkable feat here is that our approach was able to explore and exploit this inherent structure non-intrusively. Moreover, the estimates for the mean and variance of the output of interest are $\mathbb{E}[Q_{\text{hi-fi}}] \approx 0.7530$ MW and $\text{Var}[Q_{\text{hi-fi}}] \approx 0.2571$ MW$^2$, respectively.

To put into perspective the cost reduction in terms of high-fidelity simulations, a standard full tensor grid-based method with only three points per dimension, e.g., one for the center of the domain and two other for the extrema entails a total of $3^8 = 6561$ high-fidelity simulations. While this number might not be considered high for model problems, it requires roughly 53 million core-hours for the scenario under consideration, which is computationally prohibitive. In contrast, our approach required about 460,000 core-hours in total, i.e., a factor of 115 less in comparison, which was computationally feasible. In general, in large-scale applications it is unrealistic to perform more than a handful of runs, which is in par with what our method typically requires.

To better understand how our algorithm explored the 8D input space, Fig. 4 plots all pair-wise two-dimensional projections of the 57 sparse grid points obtained at the end of the refinement process. Notice that the projections involving either $\omega_{T_e}$ or $\omega_{n_e}$—the two most important input parameters—contain the most grid points, showing that indeed these directions have been explored more extensively by our method. In contrast, the projections involving the unimportant parameters, especially the two least important parameters in the considered scenario, $\hat{s}$ and $Z_{\text{eff}}$, contain the least amount of grid points.

Performing UQ and SA in realistic turbulent transport simulations in fusion devices is relevant since in virtually all experiments, it is generally very difficult to robustly explain the observed behavior and to pinpoint the most important parameters. Our method, in contrast, allows us to systematically consider all uncertain inputs, which makes such an analysis more robust. Furthermore, our algorithm can explore the entire parameter space, including regions that are not accessible by current experiments when seeking to, e.g., optimize turbulent transport. In general, UQ and SA are fundamental tasks for many real-world simulations, which can be performed at scale via our framework.

**An efficient surrogate model for the input-to-output of interest mapping.** Our method intrinsically provides an interpolation-based surrogate model of the parameters-to-(time-averaged)-heat flux mapping. To ascertain its efficiency in the scenario under consideration, we draw 32 pseudo-random test samples from the eight-dimensional input uniform distribution and use them to evaluate both the high-fidelity model provided by GENE and the obtained sparse grid interpolation surrogate. Note that the large computational cost of the high-fidelity model prohibits using a large number of test samples.

Figure 5 compares the high-fidelity results with the predictions obtained using our surrogate model, denoted by $Q_{\text{SG}}$[MW]. Notice first that the values of the heat fluxes at the 32 test samples vary broadly, from roughly 0.1–2.6 MW, indicating that these

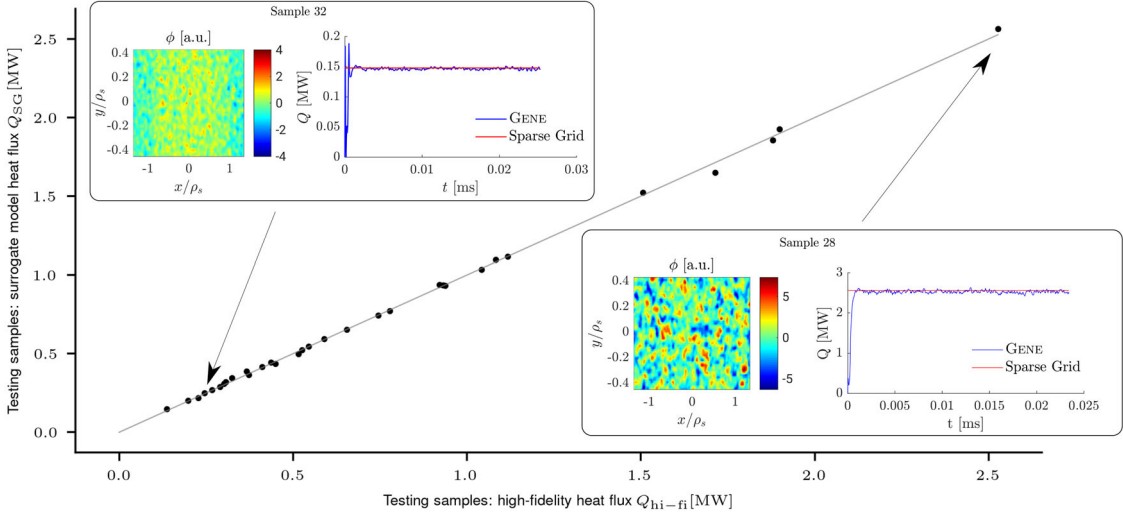

**Fig. 5 High-fidelity heat fluxes $Q_{hi-fi}$ versus heat fluxes $Q_{SG}$ predicted by the sparse grid interpolation surrogate at 32 pseudo-random test samples.** The surrogate model is accurate for both high- and low-flux values. For a more detailed visualization, we also plot the high-fidelity heat flux (on the left) as well as the time trace of the simulated heat flux (on the right), depicted in blue, and the prediction obtained with the sparse grid surrogate model, depicted in red, for two samples: #32 at low flux and #28 at high flux.

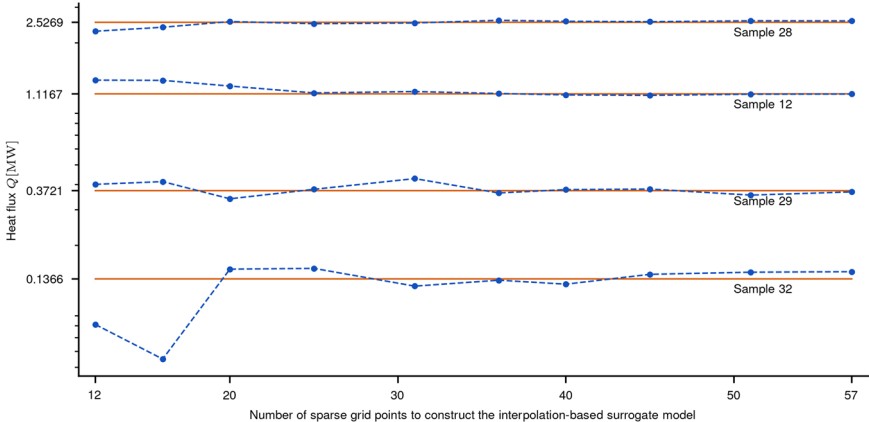

**Fig. 6 Predicted heat fluxes for four test samples.** The number of sparse grid points used to construct the surrogate model increases from 12 to 57. The four samples were chosen such that two correspond to low-flux values (#32 and #29) and two correspond to high-flux values (#12 and #28). The dashed dotted lines plot the predicted heat-flux values (i.e., $Q_{SG}$) obtained using our sparse grid surrogate model. For reference, the corresponding high-fidelity heat-flux values (i.e., $Q_{hi-fi}$) are visualized as well using solid lines.

samples are well distributed in the input domain; for a more comprehensive visualization, we also depict two examples—one of a low and the other of a high ETG heat flux. We see that the predictions yielded by the surrogate model closely match the high-fidelity values, suggesting that the surrogate model is accurate. To quantify its accuracy, we compute its mean-squared error (MSE) and obtain $\mathrm{MSE}(Q_{hi-fi}, Q_{SG}) = \frac{1}{32}\sum_{n=1}^{32}\left(Q_{hi-fi,n} - Q_{SG,n}\right)^2 = 3.0707 \times 10^{-4}$, which confirms that the surrogate is indeed accurate. Moreover, its average evaluation cost is $c_{SG} = 9.4046 \times 10^{-3}$ s, which is nine orders of magnitude smaller than the average evaluation cost of the high-fidelity model.

For a more detailed perspective on the prediction capabilities of the surrogate model, Fig. 6 plots the value of $Q_{SG}$ at four test samples, two for low-flux values and two for high-flux values, as the number of sparse grid points used to construct the surrogate model increases from 12 to 57. We observe that all four predictions converge toward a fixed heat-flux value. Even more, when using 36 grid points or more, the predicted heat-flux values vary insignificantly, indicating that the prediction uncertainty of

our surrogate model is small. For reference, we have also visualized the high-fidelity results, which are closely matched by the predictions of our surrogate model. We can therefore conclude that the obtained surrogate model, constructed at a cost of only 57 high-fidelity evaluations, is also very efficient.

In the context of fusion research, the intrinsically provided interpolation-based surrogate model can be used to predict profiles based on given heat-flux values. This, in turn, can enable the prediction and optimization of future devices, which represents one of the most important goals of computational plasma physics. Moreover, in a broader context, this surrogate can be used in subsequent optimization or multi-fidelity[35] studies.

## Discussion
UQ and SA are essential for obtaining accurate, predictive, and more robust numerical simulations of real-world phenomena. However, in many cases, the computational cost of a single simulation tends to be large, which renders most standard methods computationally infeasible. In the present article, we have demonstrated that these challenges can be overcome with

our recently developed sensitivity-driven dimension-adaptive sparse grid interpolation framework. This method explores and exploits the structure of the underlying problem in terms of lower intrinsic dimensionality and anisotropic coupling of the uncertain input parameters. The framework is fully non-intrusive and requires only prescribing the values of the input parameters and computing the output of interest. Furthermore, the method is generic and applicable to a broad spectrum of problems, and it can be used on a wide range of computing systems, from laptops to high-performance supercomputers.

We have demonstrated the power and usefulness of our framework in the context of fusion research. Our focus was on the challenging and practically relevant question on the nature of nonlinear turbulent transport in the edge region of tokamak devices. In a scenario with eight uncertain parameters and more than 264 million degrees of freedom, for which a single simulation required (on average) more than 8000 CPU hours on 896 compute cores on the Frontera supercomputer, our framework required a mere total of 57 high-fidelity simulations for UQ and SA. In addition, it intrinsically provided an accurate interpolation-based surrogate model of the parameters-to-output of interest mapping that was nine orders of magnitude cheaper than the high-fidelity model. Note that in the context of the simpler linear gyrokinetic simulations concerning turbulence suppression by energetic particles, recent research efforts have shown that the sensitivity-driven approach can be effectively used as a surrogate model for optimization[32] or as a low-fidelity model in a multi-fidelity study[36].

Since our method is based on globally defined interpolation polynomials, its main drawback is that it is generally not applicable to problems characterized by discontinuities or sharp gradients in the parameters-to-output of interest mapping. A possible remedy is an extension to instead using basis functions with local support, e.g., wavelets, which is subject to our ongoing research. We also note that even though dimension-adaptive algorithms have been shown to be effective and accurate in many previous studies, their accuracy can usually be ascertained only empirically in generic settings. Initial steps toward proving convergence have been done, for example, in ref. [37] for elliptic partial differential equations with finite-dimensional affine diffusion. Another goal for our future research is to equip our algorithm with robust data-driven methods to quantify prediction uncertainty without limiting its generality.

## Methods

**Sensitivity-driven dimension-adaptive sparse grid interpolation.** Let $f_{\text{hi-fi}} : \mathcal{X} \to \mathcal{Y}$ denote the underlying high-fidelity model. The domain $\mathcal{X} \subset \mathbb{R}^d$ is the set of the $d \in \mathbb{N}$ uncertain inputs $\boldsymbol{\theta} = [\theta_1, \theta_2, \dots, \theta_d]^T$ and the domain $\mathcal{Y} \subset \mathbb{R}$ is the range of the scalar-valued outputs $y = f_{\text{hi-fi}}(\boldsymbol{\theta})$. We note that the presented methodology can trivially employed for multi-variate outputs as well by, e.g., applying it to each output component separately. We make use of the following two assumptions: the set $\mathcal{X}$ of uncertain inputs has a product structure, i.e., $\mathcal{X} = \bigotimes_{i=1}^{d} \mathcal{X}_i$, and the input density, $\pi$, has a product structure as well, that is, $\pi(\boldsymbol{\theta}) = \prod_{i=1}^{d} \pi_i(\theta_i)$, where $\mathcal{X}_i$ is the image of the univariate density $\pi_i$ associated with input $\theta_i$. In other words, the $d$ uncertain parameters are assumed to be independent. However, we note that this assumption can be relaxed, by using, for example, a (possibly nonlinear) transformation that makes the inputs independent, such as a transport map[38].

Let $\boldsymbol{\ell} = (\ell_1, \ell_2, \dots \ell_d) \in \mathbb{N}^d$ denote a multi-index and let $\mathcal{L} \subset \mathbb{N}^d$ be a set of multi-indices. The $d$-variate sparse grid approximation of $f_{\text{hi-fi}}$ is defined as

$$\mathcal{U}^d_{\mathcal{L}}[f_{\text{hi-fi}}](\boldsymbol{\theta}) = \sum_{\boldsymbol{\ell} \in \mathcal{L}} \Delta^d_{\boldsymbol{\ell}}[f_{\text{hi-fi}}](\boldsymbol{\theta}), \quad (1)$$

where

$$\Delta^d_{\boldsymbol{\ell}}[f_{\text{hi-fi}}] = \sum_{\mathbf{z} \in \{0,1\}^d} (-1)^{|\mathbf{z}|_1} \mathcal{U}^d_{\boldsymbol{\ell}-\mathbf{z}}[f_{\text{hi-fi}}](\boldsymbol{\theta}) \quad (2)$$

are the so-called hierarchical surpluses, with $|\mathbf{z}|_1 = \sum_{i=1}^{d} z_i$. The $d$-variate surpluses $\Delta^d_{\boldsymbol{\ell}}[f_{\text{hi-fi}}]$ are a linear combination of $d$-variate approximations $\mathcal{U}^d_{\boldsymbol{\ell}-\mathbf{z}}[f_{\text{hi-fi}}]$ which

in turn are obtained by tensorizing $d$ one-dimensional operators $\mathcal{U}^i_{\ell_i-z_i}$:

$$\mathcal{U}^d_{\boldsymbol{\ell}-\mathbf{z}}[f_{\text{hi-fi}}](\boldsymbol{\theta}) = \left( \bigotimes_{i=1}^{d} \mathcal{U}^i_{\ell_i-z_i} \right)[f_{\text{hi-fi}}](\boldsymbol{\theta}). \quad (3)$$

The multi-index set $\mathcal{L}$ must allow the computation of all terms in the hierarchical surpluses (2). Such a multi-index set is called admissible or downward closed, i.e., it does not contain "holes".

To fully specify the sparse grid approximation (1), we need three ingredients: (i) the one-dimensional approximation operators $\mathcal{U}^i_{\ell_i}$ for $i = 1, 2, \dots, d$, (ii) the grid points with which we compute these 1D approximation, and (iii) the multi-index set $\mathcal{L}$. In our method, the underlying operation is Lagrange interpolation, implemented in terms of the barycentric formula for improved numerical stability. We note that our approach can also be employed for other approximation operations, such as spectral projection or quadrature. The interpolation points are weighted (L)-Leja points. For a continuous function $g : \mathcal{X}_i \to \mathbb{R}$, we define the univariate interpolation polynomial $\mathcal{U}^i_{\ell_i}$ associated to $\ell_i$ as:

$$\mathcal{U}^i_{\ell_i} : C^0(\mathcal{X}_i) \to \mathbb{P}_{P_{\ell_i}}, \quad \mathcal{U}^i_{\ell_i}[g] = \sum_{n=1}^{\ell_i} g(\theta_n) L_n(\theta), \quad (4)$$

where $\{\theta_n\}_{n=1}^{\ell_i}$ are weighted (L)-Leja points computed with respect to the density $\pi_i$:

$$\begin{aligned} \theta_1 &= \underset{\theta \in \mathcal{X}_i}{\arg\max} \; \sqrt{\pi_i(\theta)} \\ \theta_n &= \underset{\theta \in \mathcal{X}_i}{\arg\max} \; \sqrt{\pi_i(\theta)} \prod_{m=1}^{n-1} |(\theta - \theta_m)|, \; n = 2, 3, \dots, \ell_i, \end{aligned} \quad (5)$$

and $\{L_n(\theta)\}_{n=1}^{\ell_i}$ are Lagrange polynomials of degree $n-1$ satisfying the interpolation condition $L_n(\theta_m) = \delta_{nm}$, where $\delta_{nm}$ is Kronecker's delta function. When $\pi_i$ is a uniform density with support $[a, b] \subset \mathbb{R}$, as in our numerical experiments, we set $\theta_1 = (a + b)/2$. We employ (L)-Leja sequences for four main reasons. First, they are an interpolatory sequence, meaning that $n$ (L)-Leja points uniquely specify a polynomial of degree $n-1$. Note that (L)-Leja sequences are arbitrarily granular and therefore other growth strategies can be employed as well. Second, (L)-Leja points are hierarchical, meaning that evaluations from previous levels can be reused. In our context this means that adjacent levels differ by only one (L)-Leja point. Third, (L)-Leja sequences lead to accurate interpolation approximations[39]. These three properties make (L)-Leja sequences an excellent choice for higher-dimensional interpolation approximations. Finally, (L)-Leja points can be constructed for arbitrary probability densities.

Let $\mathcal{P}_\ell$ denote the set all $d$-variate degrees $\mathbf{p} = (p_1, p_2, \dots, p_d)$ with $0 \le p_i \le \ell_i - 1$ for $i = 1, 2, \dots, d$. In addition, denote by $\boldsymbol{\theta}_\mathbf{p} = (\theta_{p_1}, \theta_{p_2}, \dots, \theta_{p_d})$ the $d$-variate (L)-Leja point associated with a $d$-variate degree $\mathbf{p}$. The multi-variate interpolation approximation (3) associated to the multi-index $\ell$ reads

$$\mathcal{U}^d_\ell[f_{\text{hi-fi}}](\boldsymbol{\theta}) = \sum_{\mathbf{p} \in \mathcal{P}_\ell} f_{\text{hi-fi}}(\boldsymbol{\theta}_\mathbf{p}) L^d_\mathbf{p}(\boldsymbol{\theta}), \quad (6)$$

where the $d$-variate Lagrange polynomial $L^d_\mathbf{p}(\boldsymbol{\theta})$ is computed as $L^d_\mathbf{p}(\boldsymbol{\theta}) = \prod_{i=1}^{d} L_{p_i}(\theta_i)$, which follows from the independence assumption of the uncertain inputs.

To fully define the sparse grid interpolation approximation (1), we need to specify the third and most important ingredient, the multi-index set $\mathcal{L}$, which we determine online via our sensitivity-driven dimension-adaptive strategy. We note that adaptive sparse grid approximations were used in previous research as well. One of the first works in this direction[40] formulated an anisotropic sparse grid collocation method for solving partial differential equations with random coefficients and forcing terms. In addition, in ref. [41], both uniform and adaptive polynomial order refinement were used to assess the convergence of non-intrusive spectral or interpolation-based techniques.

We begin by determining the equivalent spectral projection representation of the multi-variate interpolation operators (6):

$$\mathcal{U}^d_\ell[f_{\text{hi-fi}}](\boldsymbol{\theta}) = \sum_{\mathbf{p} \in \mathcal{P}_\ell} f_{\text{hi-fi}}(\boldsymbol{\theta}_\mathbf{p}) L_\mathbf{p}(\boldsymbol{\theta}) = \sum_{\mathbf{p} \in \mathcal{P}_\ell} c_\mathbf{p} \Phi_\mathbf{p}(\boldsymbol{\theta}), \quad (7)$$

where $\Phi_\mathbf{p}(\boldsymbol{\theta}) = \prod_{i=1}^{d} \Phi_i(\theta_i)$ are orthonormal polynomial with respect to the input density $\pi$ and $c_\mathbf{p}$ are the corresponding spectral coefficients. For example, if $\pi$ is the uniform distribution, as in our numerical experiments, $\Phi_\mathbf{p}(\boldsymbol{\theta})$ are Legendre polynomials. To determine the spectral coefficients $c_\mathbf{p}$, we simply solve $\sum_{\mathbf{p} \in \mathcal{P}_\ell} c_\mathbf{p} \Phi_\mathbf{p}(\boldsymbol{\theta}_k) = \mathcal{U}^d_\ell[f_{\text{hi-fi}}](\boldsymbol{\theta}_k)$ for all (L)-Leja points $\boldsymbol{\theta}_k$ associated to the multi-index $\ell$. Once we have determined the spectral coefficients $c_\mathbf{p}$, we can rewrite the hierarchical interpolation surpluses (2) in terms of hierarchical spectral projection surpluses:

$$\Delta^d_\ell[f_{\text{hi-fi}}](\boldsymbol{\theta}) = \sum_{\mathbf{p} \in \mathcal{P}_\ell} \Delta c_\mathbf{p} \Phi_\mathbf{p}(\boldsymbol{\theta}), \quad \Delta c_\mathbf{p} = \sum_{\mathbf{z} \in \{0,1\}^d} (-1)^{|\mathbf{z}|_1} c_{\mathbf{p}-\mathbf{z}}, \quad (8)$$

with the convention $\Delta c_\mathbf{0} = c_\mathbf{0}$. We rewrite hierarchical interpolation surpluses in terms hierarchical projection surpluses (8) because the latter allow to trivially compute the desired sensitivity information which we use to drive the adaptive process. Specifically, from the equivalence between spectral projections and Sobol'

decompositions[42] introduced in[43], we have

$$
\begin{aligned}
\left\| \Delta_\ell^d [f_{\mathrm{hi-fi}}] \right\|_{L^2}^2 &= \sum_{\mathbf{p} \in \mathcal{P}_\ell} \Delta c_{\mathbf{p}}^2 \\
&= \Delta c_{\mathbf{0}}^2 + \Delta \mathrm{Var}_\ell [f_{\mathrm{hi-fi}}] \\
&= \Delta c_{\mathbf{0}}^2 + \sum_{i=1}^d \Delta \mathrm{Var}_\ell^i [f_{\mathrm{hi-fi}}] + \sum_{i,j=1}^d \Delta \mathrm{Var}_\ell^{i,j} [f_{\mathrm{hi-fi}}] + \cdots + \Delta \mathrm{Var}_\ell^{1,2,\dots,d} [f_{\mathrm{hi-fi}}],
\end{aligned}
\tag{9}
$$

where

$$
\Delta \mathrm{Var}_\ell^i [f_{\mathrm{hi-fi}}] = \sum_{\mathbf{p} \in \Delta \mathcal{J}_i} \Delta c_{\mathbf{p}}^2, \quad \Delta \mathcal{J}_i = \{ \mathbf{p} \in \mathcal{P}_\ell : \mathbf{p}_i \neq 0 \wedge \mathbf{p}_j = 0, \forall j \neq i \}, \tag{10}
$$

$$
\Delta \mathrm{Var}_\ell^{i,j} [f_{\mathrm{hi-fi}}] = \sum_{\mathbf{p} \in \Delta \mathcal{J}_{i,j}} \Delta c_{\mathbf{p}}^2, \quad \Delta \mathcal{J}_{i,j} = \{ \mathbf{p} \in \mathcal{P}_\ell : \mathbf{p}_i \neq 0 \wedge \mathbf{p}_j \neq 0, \mathbf{p}_n = 0, \forall n \neq i \wedge n \neq j \}, \tag{11}
$$

and so forth. $\Delta \mathrm{Var}_\ell^i [f_{\mathrm{hi-fi}}]$ represents the unnormalized Sobol' index associated with this input. Similarly, $\Delta \mathrm{Var}_\ell^{i,j} [f_{\mathrm{hi-fi}}]$ represents the unnormalized Sobol' index corresponding to pairs of uncertain inputs, i.e., interactions of two inputs. And so on until the unnormalized Sobol' index $\Delta \mathrm{Var}_\ell^{1,2,\dots,d} [f_{\mathrm{hi-fi}}]$ corresponding to the interaction of all $d$ uncertain parameters. Therefore, the $L^2$ norms (9) provide exhaustive sensitivity information about all $d$ uncertain inputs and all interactions in the subspace associated with $\ell$.

We compute the equivalent hierarchical spectral projection surpluses (8) and their $L^2$ norm (9) for all candidate multi-indices $\ell$ for refinement. We then use the unnormalized Sobol' indices given by the $L^2$ norms (9) to compute our refinement indicator, which is an integer $\mathbf{s}_\ell \in \mathbb{N}$, referred to as the sensitivity score. Initially, $\mathbf{s}_\ell = 0$. Since (9) comprises $2^d - 1$ unnormalized Sobol' indices in total, we prescribe $2^d - 1$ user-defined tolerances $\mathbf{tol} = (\mathrm{tol}_1, \mathrm{tol}_2, \dots, \mathrm{tol}_{2^d - 1})$ which are a heuristic for the accuracy with which we want the algorithm to explore the $d$ individual directions and all their $2^d - d - 1$ interactions. We compare the $m$th term in (9) with $\mathrm{tol}_m$ and if this tolerance is exceed, $\mathbf{s}_\ell$ is increased by one. In other words, if an individual parameter or an interaction are important in a candidate subspace for refinement – as compared to the prescribed tolerance – the sensitivity index will reflect this information. Therefore, $\mathbf{s}_\ell$ can be at most $2^d - 1$. We note that when $d$ is large, $2^d - 1$ will be prohibitively large making the computation of all $2^d - 1$ sensitivities in (9) infeasible. Nevertheless, in most applications it is unlikely that pairings beyond few, e.g., two, three parameters are important and therefore (9) can be truncated to comprise only these interactions.

We refine the multi-index with the largest sensitivity score noting that if two or more multi-indices have identical scores, we select the one with the largest variance $\Delta \mathrm{Var}_\ell [f_{\mathrm{hi-fi}}]$. Refining a multi-indices means that it is moved to the old index set $\mathcal{O}$ and all its forward neighbors $\ell + \mathbf{e}_i$ with $i = 1, 2, \dots, d$ that maintain $\mathcal{L}$ admissible are added to the active set $\mathcal{A}$. The refinement ends if either the prescribed tolerances $\mathbf{tol}$ are reached, if $\mathcal{A} = \emptyset$ or if a user-defined maximum level $L_{\max}$ is reached. For example, we used $L_{\max} = 20$ in our numerical experiments.

Upon termination, statistics such as expectation and variance of the output of interest, as well as sensitivity indices can be straightforwardly estimated as follows. Let $N \in \mathbb{N}$ denote the cardinality of $\mathcal{L}$ and let $\ell_m = (\ell_{m,1}, \ell_{m,2}, \dots, \ell_{m,d})$ denote the $m$th multi-index in $\mathcal{L}$, where $\ell_1 = (1, 1, \dots, 1)$. Furthermore, let $\mathcal{P}_{\mathcal{L}} = \left\{ \mathbf{p}_{\ell_m} := (\ell_{m,1} - 1, \ell_{m,2} - 1, \dots, \ell_{m,d} - 1) : \ell_m \in \mathcal{L} \right\}$ denote the set of multi-variate degrees of the equivalent global spectral projection basis. We can rewrite (6) as

$$
\mathcal{U}_{\mathcal{L}}^d [f_{\mathrm{hi-fi}}](\boldsymbol{\theta}) = \sum_{\mathbf{p}_{\ell_m} \in \mathcal{P}_{\mathcal{L}}} \Delta c_{\mathbf{p}_{\ell_m}} \Phi_{\mathbf{p}_{\ell_m}}(\boldsymbol{\theta}) = \sum_{m=1}^N \Delta c_{\mathbf{p}_{\ell_m}} \Phi_{\mathbf{p}_{\ell_m}}(\boldsymbol{\theta}), \tag{12}
$$

where the spectral coefficients $\Delta c_{\mathbf{p}_{\ell_m}}$ are computed analogously to (8). To simplify the notation in the following, denote $\mathcal{P}_{\mathcal{L}}^* = \mathcal{P}_{\mathcal{L}} \setminus \{ \mathbf{p}_{\ell_1} \}$. We estimate the expectation and variance of the high-fidelity model using these coefficients as[44]

$$
\mathbb{E}[f_{\mathrm{hi-fi}}] = \Delta c_{\mathbf{p}_{\ell_1}}, \quad \mathrm{Var}[f_{\mathrm{hi-fi}}] = \sum_{\mathbf{p}_{\ell_m} \in \mathcal{P}_{\mathcal{L}}^*} \Delta c_{\mathbf{p}_{\ell_m}}^2 = \sum_{m=2}^N \Delta c_{\mathbf{p}_{\ell_m}}^2. \tag{13}
$$

In addition, the first-order Sobol' sensitivity indices corresponding to individual parameters are computed as[7]

$$
S_{\theta_i} = \frac{\mathrm{Var}^i [f_{\mathrm{hi-fi}}]}{\mathrm{Var}[f_{\mathrm{hi-fi}}]}, \quad i = 1, 2, \dots, d, \tag{14}
$$

where $\mathrm{Var}^i [f_{\mathrm{hi-fi}}]$ denotes the contribution of $i$th input to $\mathrm{Var}[f_{\mathrm{hi-fi}}]$,

$$
\mathrm{Var}^i [f_{\mathrm{hi-fi}}] = \sum_{\mathbf{p} \in \mathcal{J}_i} \Delta c_{\mathbf{p}}^2, \tag{15}
$$

where $\mathcal{J}_i = \{ \mathbf{p}_{\ell_m} \in \mathcal{P}_{\mathcal{L}}^* : \mathbf{p}_{\ell_m,k} = 0, \forall k \neq i \}$. Analogously, indices corresponding to interactions of two parameters $\theta_i$ and $\theta_j$ are computed as

$$
S_{\theta_i, \theta_j} = S_{\theta_j, \theta_i} = \frac{\mathrm{Var}^{i,j} [f_{\mathrm{hi-fi}}]}{\mathrm{Var}[f_{\mathrm{hi-fi}}]}, \quad \mathrm{Var}^{i,j} [f_{\mathrm{hi-fi}}] = \sum_{\mathbf{p} \in \mathcal{J}_{ij}} \Delta c_{\mathbf{p}}^2, \quad i = 1, 2, \dots d - 1; j \geq i + 1, \tag{16}
$$

where $\mathcal{J}_{ij} = \{ \mathbf{p}_{\ell_m} \in \mathcal{P}_{\mathcal{L}}^* : \mathbf{p}_{\ell_m,k} = 0, \forall k \neq i \wedge k \neq j \}$, and so forth for all other interactions. Finally, the obtained sparse grid approximation intrinsically yields a surrogate model for the parameter-to-output of interest mapping as well.

**High-fidelity gyrokinetic simulation of plasma turbulence.** Gyrokinetic theory[25] provides an efficient description of low-frequency, small-amplitude, small-scale turbulence in strongly magnetized plasmas. Here, the fast gyromotion is removed from the equations, and electrically charged particles are effectively replaced by respective rings which move in a weakly inhomogeneous background magnetic field and in the presence of electromagnetic perturbations. This process reduces the kinetic description of the plasma from six to five dimensions (three spatial and two velocity space coordinates of the gyrocenters) and, even more importantly, removes a number of extremely small, but irrelevant spatio-temporal scales from the problem.

In gyrokinetics, each plasma species $s$ is described by a distribution function $F_s(\mathbf{X}, v_\parallel, \mu, t)$ whose dynamics is governed by the following equation:

$$
\frac{\partial F_s}{\partial t} + \dot{\mathbf{X}} \cdot \nabla F_s + \dot{v}_\parallel \frac{\partial F_s}{\partial v_\parallel} = \mathcal{C}. \tag{17}
$$

Here, $\mathbf{X}$ is the gyrocenter position, $v_\parallel$ is the velocity component parallel to the background magnetic field $\mathbf{B} = B\mathbf{b}$, and $\mu$ is the magnetic moment (a conserved quantity in the collisionless limit). $\mathcal{C}$ denotes a collision operation describing inter- and intra-species interactions. In our numerical experiments, we used a linearized Landau-Boltzmann collision operator. The corresponding equations of motion for a gyrocenter of a particle with mass $m$ and charge $q$ read

$$
\dot{\mathbf{X}} = v_\parallel \mathbf{b} + \frac{B}{B_\parallel^*} (\mathbf{v}_{\nabla B} + \mathbf{v}_\kappa + \mathbf{v}_E), \tag{18}
$$

$$
\dot{v}_\parallel = -\frac{\dot{\mathbf{X}}}{m v_\parallel} \cdot \left( \mu \nabla B + q \nabla \bar{\phi} \right) - \frac{q}{m} \dot{\bar{A}}_\parallel, \tag{19}
$$

where $\mathbf{v}_{\nabla B} = (\mu/(m\Omega B)) \mathbf{B} \times \nabla B$ is the grad-B drift velocity, $\mathbf{v}_\kappa = (v_\parallel^2/\Omega)(\nabla \times \mathbf{b})_\perp$ is the curvature drift velocity, and $\mathbf{v}_E = (1/B^2) \mathbf{B} \times \nabla(\bar{\phi} - v_\parallel \bar{A}_\parallel)$ is the generalized $\mathbf{E} \times \mathbf{B}$ drift velocity. Here, $\Omega = qB/m$ is the gyrofrequency, and $B_\parallel^*$ is the parallel component of the effective magnetic field $\mathbf{B}^* = \mathbf{B} + \frac{B}{\Omega} v_\parallel \nabla \times \mathbf{b} + \nabla \times (\mathbf{b} \bar{A}_\parallel)$. Finally, $\bar{\phi}$ and $\bar{A}_\parallel$ are the gyroaveraged versions of the electrostatic potential and the parallel component of the vector potential, which are self-consistently computed from the distribution function. Assuming a static background distribution function $F_{B,s}(\mathbf{X}, v_\parallel, \mu)$, which allows for the decomposition $F_s(\mathbf{X}, v_\parallel, \mu, t) = F_{B,s}(\mathbf{X}, v_\parallel, \mu) + f_s(\mathbf{X}, v_\parallel, \mu, t)$, $\phi$ can be calculated via the Poisson equation, which—expressed at the particle position $\mathbf{x}$—reads

$$
\nabla_\perp^2 \phi(\mathbf{x}) = -\frac{1}{\epsilon_0} \sum_s q_s n_{1,s}(\mathbf{x}) = -\frac{1}{\epsilon_0} \sum_s \frac{2\pi q_s}{m_s} \int B_\parallel^* f_s(\mathbf{x}, v_\parallel, \mu) dv_\parallel d\mu, \tag{20}
$$

while $A_\parallel$ is obtained by solving the parallel component of Ampère's law for the fluctuation fields:

$$
-\nabla_\perp^2 A_\parallel(\mathbf{x}) = \mu_0 \sum_s j_{\parallel,s}(\mathbf{x}) = \mu_0 \sum_s \frac{2\pi q_s}{m_s} \int B_\parallel^* v_\parallel f_s(\mathbf{x}, v_\parallel, \mu) dv_\parallel d\mu. \tag{21}
$$

Expressions for connecting $f_s(\mathbf{X}, v_\parallel, \mu)$ and $f_s(\mathbf{x}, v_\parallel, \mu)$ can be found in the literature[25]. Note that in these formulas, the time dependence has been suppressed for simplicity.

Our high-fidelity model of plasma turbulence is the Eulerian gyrokinetic code GENE[27], which solves the coupled system of Eqs. (17), (20) and (21) on a fixed grid in five-dimensional phase space using a mix of spectral, finite difference, and finite volume methods. In order to adapt the simulation volume to the dominant influences of the background magnetic field, GENE employs a field-aligned[45] coordinate system $(x, y, z)$: $x$ and $y$ are the two directions perpendicular to the magnetic field whereas $z$ parametrizes the position along $\mathbf{B}$.

When simulating ETG turbulence, the scale separation between turbulent and ambient characteristic spatial lengths is usually large, allowing us to simulate only a relatively small volume around a given field line. This is the so-called flux-tube limit, which is widely used in studies of plasma turbulence, including the present work. Variations of background fields in the direction perpendicular to the magnetic field across the simulation domain are neglected and replaced by constant gradients. Furthermore, periodic boundary conditions are used for both perpendicular directions $x$ and $y$, which are therefore numerically implemented with spectral methods (we will refer in the following to the corresponding Fourier modes as $k_x$ and $k_y$).

The magnetic geometry used for all our simulations has been constructed modeling the DIII-D tokamak, using a Fourier decomposition of the magnetic surface at $\rho = 0.95$ of an experimental magnetohydrodynamic equilibrium. This allows us to specify the values of the safety factor $q$ (the number of toroidal transits per single poloidal transit a given field line winds around the torus) and magnetic shear $\hat{s} = (x/q)(dq/dx)$ independently from the flux surface shape, which is kept fixed in all our numerical experiments. Plasma profiles are chosen representative of a pedestal. The absolute value of density and temperature $n_e$ and $T_e$, as well as the corresponding normalized inverse scale-lengths $\omega_{T_e} = -(R_0/T)(dT/dx)$ and $\omega_{n_e} = -(R_0/n)(dn/dx)$, have been considered uncertain. Here $R_0 = 1.6$ m indicates the major radius of the tokamak. Collisionality is computed consistently with the plasma parameters, and we also include a non-unitary $Z_{\mathrm{eff}} = \sum_i n_i Z_i^2 / n_e$, where the sums are over all ion species, to account for impurities. Similarly, the consistent value of $\beta_e = 2\mu_0 n_e T_e / B_0^2$ is adopted to describe electromagnetic

fluctuations. Finally, we have considered a deuterium-electron plasma and assumed an adiabatic response for the ions, i.e., $n_{1,i} = -n_e q_i \phi/T_i$, $j_{\parallel,i} = 0$ while retaining a non-unitary value of $\tau = Z_{eff}T_e/T_i$, such that we need to simulate only the evolution of the electron distribution function.

All high-fidelity runs have been carried out using a box with $n_{k_x} \times n_{k_y} \times n_z \times n_{v_\parallel} \times n_\mu = 256 \times 24 \times 168 \times 32 \times 8$ degrees of freedom. Moreover, we set $k_{y,min}\rho_s = 7$ and $L_x \simeq 2.7\,\rho_s$, where $\rho_s = c_s/\Omega$ is the ion sound radius and $c_s = \sqrt{T_e/m_i}$ is the ion sound speed. In velocity space, a box characterized by $-3 \le v_\parallel/v_{th} \le 3$ and $0 \le \mu T/B \le 9$ has been used. The employed GENE grid resolution ensures that the underlying simulations—including runs for the extrema of the parameter space, which yield the smallest and largest turbulent transport levels—are sufficiently accurate.

For a given set of input parameters $\{n_e, T_e, \omega_{n_e}, \omega_{T_e}, q, \hat{s}, \tau, Z_{eff}\}$, GENE explicitly evolves in time $f_e$ (considering a local Maxwellian for $F_B$) allowing turbulence to fully develop until reaching a quasi-steady-state. The steady state of the system is simulated long enough to collect sufficient statistics, typically for a few $R_0/c_s$ units. Throughout the simulation, the turbulent heat flow is evaluated as

$$Q(t) = \int_{\partial S} \mathbf{q}(t) \cdot \nabla x \; d\Sigma, \qquad (22)$$

where $d\Sigma$ is the surface element of the surface $\partial S$, in our case the flux surface at $\rho = 0.95$, and $\mathbf{q}(t)$ is the instantaneous energy flux induced by the generalized $\mathbf{E} \times \mathbf{B}$ advection, i.e., accounting for both electrostatic and electromagnetic perturbations:

$$\mathbf{q} = \int \frac{1}{2}mv^2 \mathbf{v_E} f_e d^3 v. \qquad (23)$$

Simulated fluxes are averaged over the quasi-steady-state phase of the run and the result provides the high-fidelity output of interest $Q_{hi\text{-}fi}$ in our numerical experiments. The value of $Q_{hi\text{-}fi}$ is computed in terms of physical units because that is ultimately the physical quantity that is relevant for practical applications. We note however that the input parameters in GENE are normalized using parameters that include also temperature and density, which are uncertain in our scenario. It is therefore necessary to account for that and show results in dimensional units as otherwise the results would be incorrect. Hence, the value of $Q_{hi\text{-}fi}$ is converted to physical (S.I.) units prior to be being used in our sparse grid algorithm.

## Data availability

The data to reproduce our results are available at https://github.com/ionutfarcas/general-uq-framework.

## Code availability

The implementation of the sensitivity-driven dimension-adaptive sparse grid approach is publicly available at https://github.com/ionutfarcas/sensitivity-driven-sparse-grid-approx. The codes to reproduce our results are publicly available at https://github.com/ionutfarcas/general-uq-framework. In addition, the GENE code used for the nonlinear gyrokinetic simulations is available at https://genecode.org/.

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

## Acknowledgements
All three authors have been supported in part by the Exascale Computing Project (No. 17-SC-20-SC), a collaborative effort of the U.S. Department of Energy Office of Science and the National Nuclear Security Administration. The authors also gratefully acknowledge the computing and data resources provided by the Texas Advanced Computing Center at The University of Texas at Austin (https://www.tacc.utexas.edu).

## Author contributions
I.-G.F. and G.M. performed research and data post-processing, including generating the figures used in the paper. F.J. supervised and directed the project. All the authors participated in the writing and revision of the manuscript.

## Competing interests
The authors declare no competing interests.
