## [Peer Review File · Communications Engineering]

Reviewers' comments:

Reviewer #1 (Remarks to the Author):

This paper presents an adaptive sparse grid interpolation method to perform uncertainty analysis. The adaptivity is based on refining the sparse grid along input dimensions where the response quantity shows high sensitivity. The authors demonstrate the approach on a computationally expensive application: turbulent transport in magnetic fusion devices. The problem has 8 input parameters. The output of interest for this application is time-averaged electron heat flux across a magnetic surface. The authors demonstrate that their reduced order model, a sparse grid interpolation based on Leja points and adaptively updated using a total of 57 high-fidelity simulations, is 9 orders of magnitude cheaper to run than the full high fidelity GENE simulation. This is an excellent result and likely will be of broad interest. I recommend the paper for publication pending minor revisions.

Overall, the approach presented is reasonable and appropriate. I would note that this is not a brand-new algorithm: adaptive sparse grid methods have been in the literature for many years. In addition to the papers cited, I might recommend a few additional citations, the first one providing theoretical underpinnings and the second one showing performance demonstrations of various adaptive approaches on test cases.

- F. Nobile, R. Tempone, and C. G. Webster. "An Anisotropic Sparse Grid Stochastic Collocation Method for Partial Differential Equations with Random Input Data." *SIAM Journal on Numerical Analysis*, 2008, Vol. 46, No. 5 (2008) <https://doi.org/10.1137/070680540>
- M. Eldred and L. Swiler. "Towards Goal-Oriented Stochastic Design Employing Adaptive Collocation Methods." Paper AIAA2010-9125, 13th AIAA/ISSMO Multidisciplinary Analysis Optimization Conference. <https://doi.org/10.2514/6.2010-9125>

The authors use essentially the same approach presented in reference 13, the JCP 2020 paper which uses Sobol' decompositions to develop a sensitivity scoring system that is employed in the dimension-adaptive process. It was helpful to have that paper attached to the package as part of the review. This paper uses that same dimension-adaptive process based on sensitivity scores: the application is different, however, and that is what makes the paper a unique contribution.

I would ask that the authors cite or list the particular software implementation they are using or building upon to implement the sensitivity scores. There are also many software libraries implementing variations of sparse grid adaptativity such as <https://tasmanian.ornl.gov/>, <https://www.uqlab.com/>, or <https://dakota.sandia.gov/>. It was not clear if the implementation is a standalone one or leverages existing tools.

I found the exposition in the paper to be a bit choppy, but I think that is mainly because I am familiar with computational journals in which the methods and algorithms are presented first, followed by the results. The style of *Communications Engineering* presents the results first, followed by the methods. I am fine with the exposition but ask that the authors address the comments below.

Specific comments:

P. 4. You make the comment: "Specifically, we determine the sensitivity information – with respect to the output of interest – of all uncertain inputs in each candidate subspace for refinement." However, variance-based decomposition and the use of Sobol' indices are not mentioned until later. Somewhere on page 4 and in the presentation of Figure 3, it should be mentioned that sensitivity scores are calculated with respect to a contribution to output variance.

p. 4. This terminology has the potential to be confusing: "We distinguish here between sensitivities corresponding to individual inputs, referred to as local sensitivity indices or indicators, and sensitivity indices associated to interactions of inputs." In most SA literature, sensitivities corresponding to

individual inputs are called "main indices" or "first order" indices, and those that include interactions are called "total indices" or "total order indices." Local sensitivities usually refer to derivatives at a particular input location. The authors are using a global, variance-based approach, thus I would recommend changing the use of the word local.

p. 4 Cleared should be clearly in: "a situation in which UQ and SA are cleared called"

p. 5. Is there any justification for varying the first two parameters +/- 10% and the rest +/-20%? Uncertainty characterization is always challenging, but this comment seems like these bounds were arbitrarily chosen. Any experimental or other evidence would be helpful.

p. 6, Figure 3: as above, the explanation of "local" indices could be confusing when these are global Sobol' indices.

p. 10. Add "infeasible" after (8) in: "making the computation of all $2d-1$ sensitivities in (8)."

Please mention software tools used for the dimension-adaptive sparse grid interpolation.

Reviewer #2 (Remarks to the Author):

The authors consider an important problem: quantifying uncertainties in predictions made with physics-based models. The key novelty is a structure-exploiting refinement strategy for training a sparse surrogate model. The authors show that their approach scales to realistic simulations and reduces the costs of uncertainty quantification from millions of core hours to a few hundred thousand. The strength of the work clearly is the demonstration at scale and it is very likely that the presented results transfer to other large-scale simulations.

The following questions remain:

1. The authors consider a test set of 32 data samples to provide empirical evidence that the obtained surrogate model is accurate on p.8. Thus, a total of 32 + 57 high-fidelity solves are required. Is there a different indicator that gives empirical evidence about the accuracy of the surrogate model, instead of requiring a test set? The authors note that the refinement indicator is a heuristic for the accuracy on p.10. Is there a convergence theory that helps bounding the error after x many refinements?
2. To understand how well the refinement strategy selects points at which to evaluate the high-fidelity model for surrogate model training, it would help to visualize them and to see which regions of the parameter domain are sampled most. The authors should provide a dimension-wise plot (projections on two dimensions) to show the selected points.
3. The authors should describe in more detail what the impact is of making tractable uncertainty quantification in this fusion simulation on the science. Is there are physical phenomenon that can be explained/understood better now that mean/variance/sensitivites can be estimated for such a large-scale simulation?

Reviewer #3 (Remarks to the Author):

The manuscript titled "A general framework for quantifying uncertainty at scale and its application to fusion research" by Farcas, Merlo, and Jenko describes a previously-developed sensitivity-driven dimension-adaptive sparse grid interpolation method for quantifying the effect of uncertain inputs on simulation outputs. The utility of the method is demonstrated by applying it to gyrokinetic simulations

of the ETG microinstability and assessing sensitivity of electron heat flux with respect to input parameters and arriving at an interpolation-based reduced model of the resulting parameters-to-heat-flux mapping. Through specialized adaptive sampling of the input parameter space, automated elimination of parameter-space dimensions that don't affect the output, and tracking of which parameter interactions are important, the method facilitates uncertainty quantification and sensitivity analysis in a way that is computationally tractable for gyrokinetic applications for which such analysis would otherwise be intractable. The method generalizes to other high-dimensional parameter space applications.

The manuscript is well-written and the general topic -- uncertainty quantification and sensitivity analysis for computationally-costly simulations (in this case applied to fusion tokamak physics) -- is at the intersection of physics and engineering and has applicability to other engineering disciplines and is therefore appropriate for the Communications Engineering journal. That said, the manuscript has a fundamental shortcoming in that it lacks originality. The method, uncertainty quantification and sensitivity analyses for high-dimensional input parameter spaces, computational cost savings, interpolation-based approximation to parameter-to-output map -- all in the context of gyrokinetic tokamak edge transport physics as modeled by the GENE code for fusion applications, have already been described in detail in previous publications (Farcas et al Journal of Computational Physics 2020, Farcas et al. Nuclear Fusion 2021). As such, the manuscript content does not constitute a novel contribution and I cannot recommend publishing this work in Communications Engineering. More detailed comments are below.

Major Comment:

1) Previous publications (Farcas et al Journal of Computational Physics 2020, Farcas et al. Nuclear Fusion 2021) already cover the material presented in this manuscript. The distinction between these previous papers and the present manuscript (e.g. JET-experiment parameters vs DIII-D-experiment parameters, ITG vs ETG instability, growth rate vs heat flux, 21 vs. 12 vs 8 uncertain parameters, different input parameter values, different numbers of input parameters, 250 vs 57 high-fidelity simulations, method vs framework) are largely cosmetic and involve simple changes to the simulation setup and/or tracking different outputs (and perhaps modified tolerances). Such modifications are simple to make, do not change how the method is applied, do not change the conclusions, and are insufficiently novel to justify publication. Discussion of the adaptive sparse-grid interpolation method/framework, application of the method to gyrokinetic tokamak-edge GENE simulations with uncertain inputs, computational cost savings for UQ and SA analysis of many-dimension parameter spaces, generalizability, have already been described in prior publications. Likewise an interpolation-based reduced model of the parameters-to-output mapping (i.e. one that can be used for optimization) is already implemented in the 2021 paper, although the term "reduced" is not used in that context.

Many of the statements/claims in the present manuscript are entirely applicable to the previous two publications: e.g.

-- "The novelty here is that, as it will be demonstrated in the following, our structure-exploiting method enables UQ and SA at

scale, which goes beyond what most existing methods offer.";

-- "to demonstrate capabilities of our sensitivity-driven approach, we employ it to study turbulent transport in magnetic confinement devices";

-- "our main goal in the following is to show that our sensitivity-driven approach enables an efficient UQ and SA in these simulations which would otherwise be impossible with standard methods";

-- "In the present article, we have demonstrated that these challenges can be overcome with the help of our recently formulated sensitivity-driven dimension-adaptive sparse grid interpolation framework, which enables UQ and SA at scale.";

-- "Here, we have demonstrated the power and usefulness of our framework in the context of magnetic confinement fusion research.";

-- ``The present study was the first of its kind, applying multi-dimensional UQ and SA to first-principles based turbulence simulations of fusion plasmas" .

2) It seems like the use of dimensional input parameters and dimensional simulation outputs can affect the results of the sensitivity analysis by suggesting that some parameters do/don't have a strong sensitivity simply based on their dimensional magnitude. For example, in the study of physics described by advection-type partial differential equations, it's often important to non-dimensionalize first before determining which terms dominate. Would the user-specified tolerances be affected by the dimensional units used? It would be helpful, if the manuscript explained the choice of using dimensional parameters rather than dimensionless parameters, and whether and how this might affect the results.

3) pg 12 ``convergence tests have been performed considering in particular all the extrema of the parameter space explored." It is not clear what a ``convergence test" means in this context. What specifically was varied and what converged? Does the interpolation-based reduced model depend on simulation resolution? Other than computational cost, what informed the choice of simulation resolution?

Additional comments:

4) pg 7 Fig 4: it's not clear how the turbulent fluctuation contour plots support the results/conclusions of the paper.

5) Minor typos / grammatical errors:

pg 3 typo: `` $3^2-1 = 7$ "

pg 4 typo: ``cleared called for"

pg 8 typo: ``Note note that in the context..."

A general framework for quantifying uncertainty at scale and its application to fusion research

Author's response to reviewer comments

Authors' Response: The authors would like to thank the reviewers for taking the time to carefully check and comment this manuscript. The comments have helped to improve the manuscript in many places.

Besides the reviewers' comments, a few minor corrections and typos in the earlier manuscript have been reported to the author and have been fixed in the present manuscript. All changes are highlighted in red in the attached manuscript.

1 Referee 1

Overall, the approach presented is reasonable and appropriate. I would note that this is not a brand-new algorithm: adaptive sparse grid methods have been in the literature for many years. In addition to the papers cited, I might recommend a few additional citations, the first one providing theoretical underpinnings and the second one showing performance demonstrations of various adaptive approaches on test cases.

Authors' Response: We thank the reviewer for pointing out these missing references. We added them in section *Sensitivity-driven adaptive refinement in Methods* in the revised manuscript.

P. 4. You make the comment: "Specifically, we determine the sensitivity information – with respect to the output of interest – of all uncertain inputs in each candidate subspace for refinement." However, variance-based decomposition and the use of Sobol' indices are not mentioned until later. Somewhere on page 4 and in the presentation of Figure 3, it should be mentioned that sensitivity scores are calculated with respect to a contribution to output variance.

Authors' Response: We adjusted the text accordingly throughout the manuscript as well as in the caption of Figure 1 (we assume and the reviewer referred to Figure 1 instead of Figure 3). We note that we have moreover shortened Figure 1's caption such that it does not exceed 350 words.

p.4. This terminology has the potential to be confusing: "We distinguish here between sensitivities corresponding to individual inputs, referred to as local sensitivity indices or indicators, and sensitivity indices associated to interactions of inputs." In most SA literature, sensitivities corresponding to individual inputs are called "main indices" or "first order" indices, and those that include interactions are called "total indices" or "total order indices." Local sensitivities usually

refer to derivatives at a particular input location. The authors are using a global, variance-based approach, thus I would recommend changing the use of the word local.

Authors' Response: We thank the reviewer for spotting this inconsistency! We amended the text accordingly throughout the manuscript.

p. 4. Cleared should be clearly in: “a situation in which UQ and SA are cleared called”

Authors' Response: Modified accordingly in the revised manuscript.

p. 5. Is there any justification for varying the first two parameters $\pm 10\%$ and the rest $\pm 20\%$? Uncertainty characterization is always challenging, but this comment seems like these bounds were arbitrarily chosen. Any experimental or other evidence would be helpful.

Authors' Response: We agree that characterizing uncertainty is challenging, especially when dealing with complex simulations. In plasma turbulence simulations, most input parameters characterize either the complex magnetic configuration or the radial density/temperature profiles (including their gradients). Experimental uncertainties exist, of course, but tend to be hard to quantify reliably. The values used in our work ($\pm 20\%$) correspond to typical values that are commonly used when dealing with turbulent transport in fusion plasmas. The larger bounds for the inverse scale-lengths compared to the to the lower, $\pm 10\%$ bounds used for their respective local values reflects the fact that the respective measurements are not directly available but must be computed from the measured profiles. We added a summary of this argument when describing the eight considered uncertain inputs and in the caption of Table 1.

p. 6. Figure 3: as above, the explanation of “local” indices could be confusing when these are global Sobol' indices.

Authors' Response: Fixed accordingly.

p. 10. Add “infeasible” after (8) in: “making the computation of all $2^d - 1$ sensitivities in (8).”

Authors' Response: Fixed accordingly.

Please mention software tools used for the dimension-adaptive sparse grid interpolation.

Authors' Response: We used our own implementation of sensitivity-driven dimension-adaptive sparse grid interpolation, which is publicly available on github. We added this information in the third paragraph in the *A framework for uncertainty propagation and sensitivity analysis in large-scale simulations* section.

2 Referee 2

1. The authors consider a test set of 32 data samples to provide empirical evidence that the obtained surrogate model is accurate on p.8. Thus, a total of $32 + 57$ high-fidelity solves are required. Is there a different indicator that gives empirical evidence about the accuracy of the surrogate model, instead of requiring a test set? The authors note that the refinement indicator is a heuristic for the accuracy on p.10. Is there a convergence theory that helps bounding the error after x many refinements?

Authors' Response: This is an excellent and important point! Empirically, the accuracy of such a reduced model can be evaluated as follows. Given a test sample, it is useful to, e.g., visualize the

predictions of the reduced model for increasing number of sparse grid point. In this way, the convergence of these predictions can be empirically ascertained, as well as their associated uncertainty. We added such a plot in the revised manuscript. Figure 6 plots four such predictions, two for low flux and two for high-flux values. Therein, we observe that as the number of grid points increases, the predicted values of the heat flux converge to a fixed value. We therefore see that the prediction uncertainty of the model is small. Another way to look at this would be to plot the spectral coefficients (which are used to compute the mean, variance and sensitivity indices) and observe their decay: a fast decay indicates a (relatively) smooth mapping, for which an interpolation or spectral projection-based approximation is usually accurate. As a side note, this makes the employed refinement indicator a good heuristic indeed since the sensitivity scores count how many unnormalized sensitivity indicators, i.e., summations of spectral coefficients exceed the imposed tolerances in each candidate subspace for refinement. Therefore, when the algorithm terminates, all these summations have become smaller than the imposed tolerances. Some first steps in the direction of proving convergence were made recently (for example, in *On the convergence of adaptive stochastic collocation for elliptic partial differential equations with affine diffusion*, SIAM Journal on Numerical Analysis, 2022 by Eigel et al., where the convergence of the dimension-adaptive algorithm was proved for elliptic problems with affine diffusion coefficients). Nevertheless, a convergence theory for generic settings, which are the target of our algorithm, would be quite difficult. In our future research, we seek to equip our strategy with robust data-driven methods to quantify prediction uncertainty without limiting its generality. We added a summary of these ideas at the of the *Discussion* section.

2. To understand how well the refinement strategy selects points at which to evaluate the high-fidelity model for surrogate model training, it would help to visualize them and to see which regions of the parameter domain are sampled most. The authors should provide a dimension-wise plot (projections on two dimensions) to show the selected points.

Authors' Response: We visualized all two-dimensional projections in the newly added Figure 4. We also added a paragraph that summarizes the new figure. As expected from the obtained sensitivity results, we observed that indeed more points were added in the important directions and interactions thereof.

3. The authors should describe in more detail what the impact is of making tractable uncertainty quantification in this fusion simulation on the science. Is there are physical phenomenon that can be explained/understood better now that mean/variance/sensitivites can be estimated for such a large-scale simulation?

Authors' Response: We thank the reviewer for pointing this out. Performing UQ and SA systematically in realistic turbulent transport simulations in fusion devices is relevant since in virtually all experiments, it is generally very difficult to robustly explain the observed behaviour and to pinpoint the specific parameters that are the most important. This is mainly due to the large experimental error bars and the typically large number of parameters. Oftentimes, experiments seek to isolate the effect of specific changes in the properties of the plasma. This, however, can result in (minor) variations in almost all parameters, making it difficult to explain the chain of causation. Our algorithm, in contrast, by systematically considering all uncertain inputs, makes such an analysis more robust and can therefore better explain experimental observations. In addition, our algorithm explores the entire parameter space, including regions that cannot be accessed

by current devices when seeking to, e.g., optimize turbulent transport. As we have shown in the paper, when the underlying problem has structure, our algorithm has the capability to perform this exploration efficiently. Finally, the reduced model obtained intrinsically via our algorithm can be used to predict profiles based on given heat flux values, which can allow predicting and optimizing future devices, which is one of the most important goals of computational plasma physics. We have added these points in the revised manuscript at the end of section *Accurate uncertainty propagation and sensitivity analysis at a cost of only 57 high-fidelity simulations* and in the beginning of section *An efficient reduced model for the parametrized input-output mapping*.

3 Referee 3

1) Previous publications (Farcas et al Journal of Computational Physics 2020, Farcas et al. Nuclear Fusion 2021) already cover the material presented in this manuscript. The distinction between these previous papers and the present manuscript (e.g. JET-experiment parameters vs DIII-D-experiment parameters, ITG vs ETG instability, growth rate vs heat flux, 21 vs. 12 vs 8 uncertain parameters, different input parameter values, different numbers of input parameters, 250 vs 57 high-fidelity simulations, method vs framework) are largely cosmetic and involve simple changes to the simulation setup and/or tracking different outputs (and perhaps modified tolerances). Such modifications are simple to make, do not change how the method is applied, do not change the conclusions, and are insufficiently novel to justify publication. Discussion of the adaptive sparse-grid interpolation method/framework, application of the method to gyrokinetic tokamak-edge GENE simulations with uncertain inputs, computational cost savings for UQ and SA analysis of many-dimension parameter spaces, generalizability, have already been described in prior publications. Likewise an interpolation-based reduced model of the parameters-to-output mapping (i.e. one that can be used for optimization) is already implemented in the 2021 paper, although the term “reduced” is not used in that context.

Many of the statements/claims in the present manuscript are entirely applicable to the previous two publications: e.g.

–“The novelty here is that, as it will be demonstrated in the following, our structure-exploiting method enables UQ and SA at scale, which goes beyond what most existing methods offer.”;

–“to demonstrate capabilities of our sensitivity-driven approach, we employ it to study turbulent transport in magnetic confinement devices”;

–“our main goal in the following is to show that our sensitivity-driven approach enables an efficient UQ and SA in these simulations which would otherwise be impossible with standard methods”;

–“In the present article, we have demonstrated that these challenges can be overcome with the help of our recently formulated sensitivity-driven dimension-adaptive sparse grid interpolation framework, which enables UQ and SA at scale.”;

–“Here, we have demonstrated the power and usefulness of our framework in the context of magnetic confinement fusion research.”;

–“The present study was the first of its kind, applying multi-dimensional UQ and SA to first-principles based turbulence simulations of fusion plasmas” .

Authors' Response: We thank the reviewer for these comments, they helped us clarify important details in the revised manuscript. In hindsight, we realize that our original text was not clear enough to avoid such misunderstandings. Therefore, we would like to explain the situation in more

detail here. The physical phenomenon at the center of the application discussed in our work is turbulent transport in fusion plasmas. The latter is a chaotic process in space and time, in which a large number of degrees of freedom (i.e., plasma micro-instabilities) are strongly coupled in a highly nonlinear fashion. In this context, *linear* simulations are often used to gain valuable insights regarding some general trends or parameter dependencies. However, to make reliable quantitative predictions, *nonlinear* simulations are needed, and their computational cost is about four orders of magnitude higher in our case. Now, while we have used linear simulations in the previous two papers, the present work is based on nonlinear simulations, therefore fully addressing - for the very first time - the problem at hand in its entirety.

The demonstration of the applicability of our UQ/SA framework is non-trivial in two ways. First, the very fact that it can be applied to real-world engineering problems simulated on large-scale supercomputers is in itself a remarkable achievement - exceeding the capabilities of conventional approaches in fundamental ways. Second, the nonlinear problem is more complex and challenging on many fronts, involving, e.g., spatio-temporal averages and much larger intrinsic uncertainties, rendering the applicability of our framework remarkable. For both of these reasons, our present work is pushing the envelope and opening doors in various directions of interest to the readers of this journal.

We admit, though, that these points have not been sufficiently emphasized in the original submission. We have clarified them in the revised manuscript: see the sections Introduction [last paragraph], Application [1st and 2nd paragraphs], Results, and Discussion.

2) It seems like the use of dimensional input parameters and dimensional simulation outputs can affect the results of the sensitivity analysis by suggesting that some parameters do/don't have a strong sensitivity simply based on their dimensional magnitude. For example, in the study of physics described by advection-type partial differential equations, it's often important to non-dimensionalize first before determining which terms dominate. Would the user-specified tolerances be affected by the dimensional units used? It would be helpful, if the manuscript explained the choice of using dimensional parameters rather than dimensionless parameters, and whether and how this might affect the results.

Authors' Response: We thank the reviewer for this observation. Related to our experiments, we note that the input parameters in GENE are normalized using parameters that include also temperature and density, which are uncertain in our scenario. Nevertheless, the output of interest (i.e. the heat flux) was computed in terms of physical units because that is ultimately the physical quantity that is relevant for practical applications. It is therefore necessary to account for that and show results in dimensional units as otherwise the results would be incorrect. We added this information at the end of section *High-fidelity gyrokinetic simulation of plasma turbulence* in *Methods*.

Generally, the important ingredient in our dimension-adaptive algorithm is the (global) sensitivity information computed in each candidate subspace for refinement. Global sensitivity analysis reflects the properties of the underlying model, therefore, whether the employed model is dimensional or non-dimensional should not affect the applicability of the method itself. Rather, the method will explore and exploit the properties of the underlying numerical model. Moreover, regarding the choice of the tolerances, in general in sparse grid approximations, the higher-order contributions tend to be smaller and smaller, which means that the quantities that are compared with the tolerances (in our case, the unnormalized sensitivities in all candidate spaces for refinement) will be smaller and smaller as well. Whether the magnitude of these quantities changes significantly

or not when the models are dimensional vs. non-dimensional should not affect the method either. In addition, as it was pointed out in the manuscript at the end of the third paragraph in section *A framework for uncertainty propagation and sensitivity analysis in large-scale simulations*, the tolerances do not need to be fixed a priori. Rather, since the approach is hierarchical, the user can start with large tolerances which can be sequentially decreased at no additional cost in terms of numerical simulations. We added a summary of this discussion at the end of the third paragraph in section *A framework for uncertainty propagation and sensitivity analysis in large-scale simulations*.

3) pg 12 “convergence tests have been performed considering in particular all the extrema of the parameter space explored.” It is not clear what a “convergence test” means in this context. What specifically was varied and what converged? Does the interpolation-based reduced model depend on simulation resolution? Other than computational cost, what informed the choice of simulation resolution?

Authors’ Response: We agree that this statement can be confusing. We clarified what was meant by “convergence tests” in the revised manuscript. By convergence we refer to the numerical convergence of the heat flux computed from the high-fidelity simulations with respect to e.g., the grid resolution used in GENE. The heterogeneity of the results of these nonlinear turbulent simulations could, in principle, require a different resolution for each parameter instance and a subsequent convergence analysis of each such simulation. However, such an approach is clearly not feasible. Numerical convergence of GENE simulations needs nevertheless to be ascertained and to do so most effectively, we exploited the fact that in the scenario under consideration only ETG modes are unstable, which implied that a sufficiently smooth dependence of the heat flux with respect to the uncertain inputs was expected. Therefore we have considered the extrema of the parameters space and for each extrema verified that GENE simulation results are converged with respect to numerical resolution in both configuration and velocity space as well as numerical dissipations. All runs have then been performed considering the set-up necessary to resolve the most demanding case. This, on the other hand, also means that a fraction of the high-fidelity simulations have over-resolved turbulence. Such a choice, albeit slightly more expensive from the computational side, is nevertheless clearly much easier to handle. We also note that as any other data-driven approach, the sensitivity-driven approach depends strongly on the GENE simulations and their resolutions. Failing to properly resolve the turbulent scales would inevitably skew the parameter space exploration. We added a summary of this discussion towards the end of section *High-fidelity gyrokinetic simulation of plasma turbulence in Methods*.

4) pg 7 Fig 4: it’s not clear how the turbulent fluctuation contour plots support the results/conclusions of the paper.

Authors’ Response: The contour plots are meant to show more details about the computed high-fidelity heat fluxes. They were not intended to support the results/conclusions of the paper. We added this clarification in the caption of the figure.

5) Minor typos / grammatical errors: pg 3 typo: “ $3^2 - 1 = 7$ ”

pg 4 typo: “cleared called for”

pg 8 typo: “Note note that in the context...”

Authors’ Response: Fixed accordingly in the revised manuscript.

REVIEWERS' COMMENTS:

Reviewer #1 (Remarks to the Author):

The authors have sufficiently addressed the reviewers' comments in this revision. They added two figures which are helpful, provided more detail and discussion about convergence, clarified the use of "sensitivity index" including Sobol' first and total effects indices, highlighted the differences between this paper and their previous papers, added a pointer to the repository for the code and examples, addressed experimental uncertainties, and made the grammatical fixes suggested. At this point, I feel the paper is ready for publication.

Reviewer #2 (Remarks to the Author):

The authors have addressed all my comments and I therefore recommend this work for publication.

Reviewer #3 (Remarks to the Author):

The authors have addressed my concerns. Despite significant overlap with material presented in Farcas et al JCP 2020 and Farcas et al NF 2021, the manuscript presents a well-written self-contained case study that may be useful to other fields that deal with UQ for computationally-costly nonlinear models. For that reason I recommend it for publication.